# Enhancement of Repeat-Mediated Deletion Rearrangement Induced by Particle Irradiation in a *RecA*-Dependent Manner in *Escherichia coli*

**DOI:** 10.3390/biology12111406

**Published:** 2023-11-07

**Authors:** Zhiyang Hou, Zelin Xu, Mengying Wu, Liqiu Ma, Li Sui, Po Bian, Ting Wang

**Affiliations:** 1Teaching and Research Section of Nuclear Medicine, School of Basic Medical Sciences, Anhui Medical University, Hefei 230032, China; houzy@mail.ustc.edu.cn (Z.H.); x86135415343121@outlook.com (Z.X.); wumengyingyx@hotmail.com (M.W.); bianpo@ahmu.edu.cn (P.B.); 2Key Laboratory of High Magnetic Field and Ion Beam Physical Biology, Hefei Institutes of Physical Science, Chinese Academy of Sciences, Hefei 230031, China; 3Science Island Branch, Graduate School of USTC, Hefei 230026, China; 4Department of Nuclear Physics, China Institute of Atomic Energy, Beijing 102413, China; lisui@ciae.ac.cn; 5National Innovation Center of Radiation Application, Beijing 102413, China

**Keywords:** repeat-mediated deletion, high LET irradiation, DSB complexity, *Escherichia coli*

## Abstract

**Simple Summary:**

Repeat-mediated deletion (RMD) is generally induced by a DNA double-strand break (DSB), but the effect of DSB complexity on RMD initiation is unclear. In order to address this issue, this research firstly used an *Escherichia coli* reporter line in which *amp* restoration was controlled by *lacI* repeats. Then, particle irradiation was used to stochastically generate complex DSBs. The results clearly confirmed the enhancement of RMD rearrangement induced by proton and carbon irradiation in a dose- and LET-dependent manner. Meanwhile, RMD rearrangement was suppressed by intermolecular homology, in which the length of homology is more important than the composition of homology. Exogenous recombinase could significantly promote particle irradiation-induced RMD events. The *RecA*-dependent pathway was suggested to be involved in the enhancement of RMD under particle irradiation. These results could broaden our understanding of RMD generation and help assess the repair process of complex DSBs located around the repeat sequences.

**Abstract:**

Repeat-mediated deletion (RMD) rearrangement is a major source of genome instability and can be deleterious to the organism, whereby the intervening sequence between two repeats is deleted along with one of the repeats. RMD rearrangement is likely induced by DNA double-strand breaks (DSBs); however, it is unclear how the complexity of DSBs influences RMD rearrangement. Here, a transgenic *Escherichia coli* strain K12 MG1655 with a *lacI* repeat-controlled amp activation was used while taking advantage of particle irradiation, such as proton and carbon irradiation, to generate different complexities of DSBs. Our research confirmed the enhancement of RMD under proton and carbon irradiation and revealed a positive correlation between RMD enhancement and LET. In addition, RMD enhancement could be suppressed by an intermolecular homologous sequence, which was regulated by its composition and length. Meanwhile, RMD enhancement was significantly stimulated by exogenous λ-Red recombinase. Further results investigating its mechanisms showed that the enhancement of RMD, induced by particle irradiation, occurred in a *RecA*-dependent manner. Our finding has a significant impact on the understanding of RMD rearrangement and provides some clues for elucidating the repair process and possible outcomes of complex DNA damage.

## 1. Introduction

Chromosome rearrangement is the main cause of genetic instability [1] and is more likely due to repetitive sequences, which are widespread in the genomes of prokaryotic and eukaryotic organisms. For a long time, genome repeat sequences have been considered as “junk DNA”; however, current evidence indicates that variations in repeats can alter the expression of genes, while changes in the number of repeats have been linked to certain human diseases [2]. The nature of genomic variation caused by repeats depends on their relative orientation, whereby repetitive sequences are classified into two types: one is direct repeats, which are oriented in the same direction, while the other is inverted repeats, which are oriented opposite each other [3,4,5]. Direct repeats result in deletions and duplications, whereas inverted repeats might cause inversions in the regions flanked by them [6,7]. In bacterial genomes, direct repeats are more common than inverted repeats, and the systematic investigation of direct sequence recombination has provided some insight into the molecular mechanisms involved in genomic rearrangement, whereby repeat-mediated deletions (RMDs) are a type of chromosomal rearrangement between two homologous sequences which causes the sequence between the repeats to be lost, along with one of the repeats [8]. In *Escherichia coli* (*E. coli*), RMD rearrangement between direct repeats occurs via both *RecA*-dependent and *RecA*-independent mechanisms, depending on the size of the repeats and on the intervening sequences between the repeats [9,10]. *RecA*-dependent recombination is the main mechanism that maintains genomic integrity, whereby the single-strand DNA (ssDNA) tracts that have formed at break sites are used to search for a homologous double-strand DNA to use as a template to repair the break [11,12], while *RecA*-independent sequence rearrangement contains three major mechanisms: simple-strand annealing, sister chromosome slipped misalignment, and single-strand annealing [13,14]. 

RMD might be generally associated with DNA breaks [15]. Each cell in a living organism is constantly exposed to various detrimental factors including physical, chemical, and biological factors which could cause dangerous damage to their genetic carrier DNA. Among those factors, radiation is a well-accepted DNA damage inducer; however, the phenomena and mechanisms of radiation-induced rearrangement of repetitive sequences have not been expansively investigated. Using the bacterium *Deinococcus radiodurans*, γ-irradiation induced a deletion of 438 bp direct repeats, which was suggested to be *RecA*-independent and in which single-strand annealing (SSA) was the driving force promoting the occurrence of DNA RMD events [14]. Whereas recombination between chromosomal and plasmid DNA has been shown to be strictly dependent on the RecA and RecF proteins, it has also been proven that the proximity of DSBs relative to repeat sequences could affect deletion rearrangement. Moreover, in a recent study, RMD rearrangement was induced with two DSBs: the 5′ DSB that was just downstream from the first repeat and the 3′ DSB that was at a considerable distance upstream of the second repeat. These data confirmed that an increase in the 3′ DSB/repeat distance from 3.3 kb to 28.4 kb only caused a modest decrease in rearrangement frequency, meaning that deletion can be induced by a chromosomal break that is located far away from a repeat [16,17]. Therefore, this finding inspired us to posit that the features of DSBs could deeply influence this event. However, compared to the proximity of a DSB, the complexity of the DSB represents a more critical feature, thus elucidating that the complexity of DSBs in RMD rearrangement will better facilitate the understanding of the mechanism involved in repeat-related rearrangements. 

DNA damage outcomes are determined by radiation quality, whereby low linear energy transfer (LET) irradiation, such as X- and γ-rays, generates low-complexity DNA damage, whereas high LET irradiation, such as protons and heavy ions, causes high-complexity DNA damage [18]. Complex damage comprises at least two DNA lesions within one helical turn of DNA, while a clustered DSBs represents the most severe form of complex DNA damage and poses a serious challenge for DNA damage repair. Additionally, its consequences are more likely to cause gross genomic rearrangement. As a special type of rearrangement, the performance of RMD under high-LET irradiation attracts our interest. Hence, by utilizing particle irradiation, such as proton and carbon irradiation, different complexities of DSBs could be generated. Therefore, this work intends to investigate the mechanisms of DSB complexity and their impact on RMD rearrangement by using an *E. coli* reporter strain, which is well established for studying the performance of repeat sequences under high LET radiation circumstances.

## 2. Materials and Methods

### 2.1. Bacterial Strains and Growth Conditions

All bacterial strains used in this study were derived from *E. coli* K12 MG1655. All cultures were incubated at 37 °C or 30 °C in Luria–Bertani (LB) medium (10 g/L, tryptone, 5 g/L yeast extract, and 10 g/L NaCl). When appropriate, antibiotics were supplemented as follows: apramycin (50 μg/mL), kanamycin (50 μg/mL), or ampicillin (50 μg/mL) [19].

### 2.2. Establishment of lacI Direct Repeat Reporter Strain and Mutant Strain of E. coli

All constructs, sgRNAs, primers, and N20 sequences followed by the PAM used in this study are presented in Appendix A. All plasmids listed in Appendix A were constructed using the Golden Gate method [20]. Plasmids and genomic DNA were extracted using the AxyPrep kit (Corning) according to the manufacturer’s instructions. DNA fragments were PCR-amplified using KOD-Plus-Neo polymerase. Restriction endonucleases BsaI and T4 DNA ligase were purchased from Thermo Scientific. The primers listed in Appendix A were designed using the web-based software J5 DeviceEditor (version number:1.0.3, URL: https://j5.jbei.org/, accessed on 23 October 2019) to create linear dsDNA products [20].

The reporter strain and deletion mutants (Δ*recA* and Δ*recB*) were constructed using a two-plasmid-based CRISPR/Cas9 genome editing system, and the recombineering method was performed according to a previously published protocol [21]. Briefly, for reporter strain construction, we inserted the sequences as the N20 sequences of guide RNA (gRNA) within the *lacZ* and *mhpR* genes, and as knock-in sequences in plasmid-encoded single-guide RNA (psgRNA) within the the *amp* and *lacI* genes, which resulted in the construction of plasmids of plasgRNA and pmlsgRNA, respectively. The MG1655 strain was first transfected with plasgRNA and pRedCas9, and 2 g/L arabinose was added to induce *Cas9* expression at 30 °C. After the knock-in strains were screened by PCR amplification and DNA sequencing, the plasmids were removed through multiple-generation cultures at 37 °C. The resulting strain was further transfected with both pmlsgRNA and pRedCas9. After conducting similar induction and screening processes, a transgenic strain harboring the *lacI:lacI:lacO::amp* gene was established and named MG1655^TR^. Similarly, deletion mutants (Δ*recA* and Δ*recB*) were induced into MG1655^TR^ using prasgRNA with the *RecA* gene and prbsgRNA with the *RecB* gene as N20 sequences. We also constructed another series of psgRNA-derived plasmids, known as ph1sgRNA, ph2sgRNA, ph3sgRNA, and ph4sgRNA, which contained overlapping homologous segments with the *lacI* gene sequence. Here, the length of homology for ph1sgRNA was 213 bp and targeted the *lacI* gene from +114 to +326; the length of homology for ph2sgRNA was 159 bp and targeted the *lacI* gene from +512 to +670; the length of homology for ph3sgRNA was 268 bp and targeted the *lacI* gene from +945 to +1212; and the length of homology for ph4sgRNA was 1212 bp, targeting the *lacI* gene from +1 to +1212. The psgRNA plasmid was used as the blank control.

### 2.3. Particle Irradiation of E. coli Strains

The irradiation experiment was performed according to a previous study. In detail, a single colony of the aforementioned *E. coli* strains was added to 5 mL of LB medium in a sterile 50 mL centrifuge tube and incubated at 37 °C with shaking at 200 rpm for 6 h, or until an OD600 value of approximately 1 × 10^8^ colony-forming units (CFUs)/mL was reached. Then, 1 mL of the culture was transferred to a 35 mm Petri dish (approximately 0.3 mm in depth) and subjected to proton or carbon irradiation. Proton irradiation was performed at the Beijing Tandem Accelerator Nuclear Physics National Laboratory, China, with an energy of 22 MeV and LET of 2.61 keV/μm. Carbon irradiation was performed at the Heavy Ion Research Facility in Lanzhou, China, using 80 MeV/nucleon with a LET of 31.46 keV/μm.

### 2.4. Measurement of E. coli Survival Rate in Response to Particle Irradiation

The irradiated bacteria (five independent samples for each irradiated treatment) were diluted and plated on LB plates, then incubated overnight at 37 °C. The number of colonies was counted to determine the CFUs. Survival rate was calculated by dividing the CFUs of the irradiated cultures by the CFUs of the culture without irradiation (0 Gy group or CT group). The final data represent the average of five independent experiments.

### 2.5. Detection of Ampicillin (Amp)-Resistant Clones

The irradiated bacteria were diluted and plated on control LB agar plates, with the selection plates containing 100 μg/mL of ampicillin (Amp), and then incubated overnight at 37 °C. CFUs were counted manually using a dissecting microscope (ES-18BZL, Motic, Xiamen, China). The Amp^R^ clones were further verified by collecting clones grown on Amp selection plates, which were then singly cultured in an LB liquid medium containing 100 μg/mL ampicillin. The mutation rate was calculated by dividing the number of Amp^R^ clones on the Amp selection plates by those on the control LB plates. The final data represent the average of five independent experiments.

### 2.6. Amplification and Sequence Analysis of lacI Gene

Verified Amp^R^ clones were collected from five independent experiments for further PCR analysis; mutant clones were suspended in 100 μL H_2_O and used as the PCR template to amplify the *lacI* gene. In the amplification reaction (total volume: 25 μL), 5 μL of the template of mutant clones (100–500 ng DNA), 0.1 units Taq DNA polymerase mix, 200 μM dNTP, and 10 pmol of each primer were added and incubation was performed according to the manufacturer’s instructions. The primers are listed in Appendix A. The PCR product for further DNA sequencing was obtained by Sangon Biotech Company (Shanghai, China). 

### 2.7. Statistical Analysis

The results for *E. coli* survival rate and mutation rates of Amp^R^ clones are presented as the mean ± standard deviation. Differences between the two groups were evaluated by Student’s *t*-test. Differences in Amp^R^ distribution were evaluated by the chi-square test using SPSS. Significance was considered at *p* < 0.05 and is indicated as * for *p* < 0.05 and ** for *p* < 0.01. 

## 3. Results

### 3.1. Enhancement of RMD Rearrangement of LacI Repeats by Particle Irradiation

To investigate RMD rearrangement under particle irradiation, *E. coli* K12 MG1655 (MG1655^TR^) was used. In this strain, an ampicillin-resistance-based reporter assay was developed. It uses two *lacI* direct repeats which are separated by a 405 bp spacer, located upstream of *E. coli lac* operon (*lacO*) site. This leads to the *Amp* gene being constitutively repressed by the regulatory *lacI* gene encoding the *lacI* repressor in combination with the *lacO* operator in the promoter. Therefore, the restoration of *Amp* resistance (Amp^R^) results from a loss of function by the *lacO* or/and two *lacI* genes. Based on the integrity of the two *lacI* gene repeats, the restoration of *Amp* resistance (Amp^R^) was classified into three types: type I implied that both *lacI* genes maintained their structural integrity together with synchronous mutations on the two *lacI* or/and on *lacO*; type II means that only one *lacI* remained due to RMD rearrangement and mutation on *lacI* or/and on *lacO*; type III was a large fragment deletion or complex structural variation that occurred on *lacI,* as schematically shown in Figure 1A. Subsequently, the MG1655^TR^ strain was subjected to different doses of proton and carbon irradiation, respectively. DSB complexity was determined by radiation doses and radiation quality. Survival rates were firstly determined and shown in Figure 1B, particle irradiation could significantly decrease the survival rate of *E. coli* in both dose-dependent and LET-dependent manners (in all cases, *p* < 0.01). Further, both particle irradiations led to an increased proportion of Amp^R^ clones compared to the control group (0 Gy) (in all cases, *p* < 0.01), also in a dose- and LET-dependent manner, as shown in Figure 1C. Next, the restoration pattern of Amp^R^ activation was explored. Firstly, 100 Amp^R^ clones generated in the 0 Gy group during five independent experiments were analyzed. It was shown that 96% of the Amp^R^ clones were classified as type I, 3% as type II, and 1% as type III. Therefore, under spontaneous DSB conditions, the activation of Amp^R^ is mainly due to small point mutations in both *lacI* gene repeats or/and *lacO*. However, particle irradiation caused a significant reduction in the proportion of type I and an increase in the proportion of type II, from 3% (3/100, CT) to 46.6% (55/118, 50 Gy proton irradiation), 51.9% (79/152, 100 Gy proton irradiation), 57.7% (45/78, 200 Gy proton irradiation), 61.1% (115/188, 50 Gy carbon irradiation), 66% (138/209, 100 Gy carbon irradiation), and 70.2% (111/158, 200 Gy carbon irradiation). The results also showed that particle irradiation tended to enhance the RMD rearrangement of *lacI* repeats in a dose- and LET-dependent manner, as shown in Figure 1D. Meanwhile, the prevalence of type III increased synchronously, especially under carbon irradiation, from 1% (1/100, 0 Gy) to 6.9% (13/188, 50 Gy), 10% (21/209, 100 Gy), and 10.7% (17/158, 200 Gy). Considering the LET of proton and carbon irradiations, an increase in types II and III might be due to the presence of more complex DNA damage. 

Here, the main classification standard for type II was the occurrence of RMD, with only one *lacI* remaining; however, the activation of the *amp* gene requires the loss of function of the remaining *lacI* or/and *lacO*, so the mutation spectrum of type II for 0 Gy and 100 Gy proton and carbon irradiation was further analyzed. The results showed that 66.7% (2/3) of Amp^R^ clones in the 0 Gy group were *lacO* mutations and base pair substitutions, while the remaining 33.3% (1/3) were located in *lacI* following a base insertion, as shown in Table 1. Analysis of 100 Gy proton irradiation revealed that 77.6% (45/58) of the Amp^R^ clones were *lacO* mutations, and the remaining 22.4% (13/58) were located in *lacI*. All the *lacO* mutations were base pair substitutions, while the *lacI* mutations were dominated by single insertions (61.5%, 8/13) followed by single base pair substitution mutations (38.5%, 5/13), as shown in Table 2. For 100 Gy carbon irradiation, 49.1% (52/106) of Amp^R^ clones were *lacO* mutations and base pair substitutions, while the remaining 50.9% (54/106) were located in *lacI*. The spectrum was dominated by single base pair substitutions (57.4%, 31/54), followed by single insertions/deletions (22.2%, 12/54) and >4 bp large fragment deletions or insertions (20.4%, 11/54), as shown in Table 3. 

Further, we attempted to determine whether the mutation in the *lacI* gene was generated alongside the occurrence of RMD. Thus, the Amp^R^ type I clones were further sequenced in an attempt to provide some clues for elucidating this issue. In type I Amp^R^ clones in the 0 Gy group, only 3.2% (3/94) of mutations were located in both *lacI* repeats, whereas their number was 12.1% (8/66) in the 100 Gy proton irradiation group and 28% (14/50) in the 100 Gy carbon irradiation group, respectively, as shown in Appendix A. Interestingly, the mutations distributed in the two *lacI* genes were totally identical, which possibly suggested that multiple base damages occurred in the *lacI* repeats, alongside the small point mutation generated during the DNA damage repair process. When homologous recombination is used to repair DSBs located in the spacer sequence between *lacI* repeats or flanking sequences, the mutant *lacI* repeat might be used as the template, which leads to the mutation being copied into the later-repaired *lacI* repeat, which is to say that in type I, the *lacI* gene might undergo an intact recombination, which would provide some clues for the time course of mutation generation in *lacI* and RMD rearrangement of *lacI*, thus suggesting that mutation in the *lacI* gene occurs prior to RMD and the possibility that the mutation might be associated with HR repair was also excluded. 

### 3.2. Suppressive Effect of Intermolecular Homology on RMD Frequency 

In the previous results, the enhancement of RMD rearrangement induced by particle irradiation mainly occurred between two intramolecular *lacI* repeat sequences. Therefore, it raised a question about the effect of intermolecular homology on radiation-induced RMD. To investigate this issue, we artificially transfected four plasmids (ph1sgRNA, ph2sgRNA, ph3sgRNA, and ph4sgRNA) harboring various lengths and compositions of *lacI* homology into MG1655^TR^ system, respectively, in addition, psgRNA which harbored no *lacI* homology was set as blank control. The schematic is shown in Figure 2A. Then, these five plasmid-harboring systems were subjected to 100 Gy of either proton or carbon irradiation. The results showed that particle irradiation decreased cell survival rates, although no significant changes were observed in the presence of intermolecular homology (*p* > 0.05, compared to psgRNA, in all cases) (Figure 2B). However, the increased mutation rates induced by particle irradiation without homology (psgRNA) were significantly changed under the presence of intermolecular homology, as shown in Figure 2C. In the MG1655^TR^ (ph2sgRNA) strain, the mutation rate was similar to that of the MG1655^TR^ (psgRNA) strain either for proton or carbon irradiation (*p* > 0.05, in both cases, compared to psgRNA). However, the mutation rate for MG1655^TR^ (ph3sgRNA) was significantly increased under carbon irradiation (*p* < 0.01) and slightly increased under proton irradiation (*p* > 0.05). Conversely, both ph1sgRNA and ph4sgRNA significantly repressed the *amp* gene activation induced by particle irradiation (in all cases, *p* < 0.01), especially for ph4sgRNA, which contained the whole length of the *lacI* homology and an extremely low mutation rate was observed (Figure 2C). 

Next, we analyzed the effect of intermolecular homology on RMD. The mutation types of Amp^R^ clones generated from five cells harboring different plasmids under particle irradiation were molecularly analyzed. For ph1sgRNA, ph2sgRNA, and ph3sgRNA, RMD frequencies in the CT group were identical, and only slightly higher than psgRNA, but with no significant difference (in all cases, *p* > 0.05). However, for ph1sgRNA, ph2sgRNA, and ph3sgRNA, enhancement trends in RMD frequency induced by proton irradiation were slightly suppressed compared to psgRNA (*p* > 0.05, in all cases), whereas they were significantly suppressed under carbon irradiation compared to psgRNA (*p* < 0.01 for ph1sgRNA and ph2sgRNA, *p* < 0.05 for ph3sgRNA) (Figure 2D). Furthermore, for ph4sgRNA, which harbored the whole *lacI* homology, enhancement trends in RMD frequency induced by particle irradiation were totally suppressed—3.3% versus 50% for proton irradiation and 4.7% versus 63.3% for carbon irradiation (*p* < 0.01, in both cases). This observed suppressive effect might be initiated by intermolecular homology which could compete with intramolecular *lacI* repeats during DSB repair. Therefore, this competition can lead to a decrease in the frequency of RMD rearrangement between intramolecular *lacI* sequences. Considering the significant differences in composition between ph1sgRNA, ph2sgRNA, and ph3sgRNA, and the differences in length between ph1sgRNA and ph4sgRNA, the competitive effect was regulated by the compositions and lengths of the intermolecular homology. 

### 3.3. Stimulative Effect of Exogenous Recombinase on RMD Initiated by Particle Irradiation

RMD rearrangement events often result from homologous recombination during the DSB repair process. Therefore, we further artificially increased cell recombination capacity to examine the effect of exogenous recombinase on RMD events induced by particle irradiation. The MG1655^TR^ strain was transfected with the pRedCas9 plasmid, which constitutively expressed λ-Red recombinase under arabinose induction (MG1655^TR^ Red^+^). Conversely, the MG1655^TR^ strain was transfected with the pRedCas9 plasmid without arabinose induction (MG1655^TR^ Red^−^) and used as the control. The schematic is shown in Figure 3A. Next, they were subjected to 100 Gy proton or carbon irradiation, respectively. The results showed that exogenous λ-Red recombinase expression (MG1655^TR^ Red^+^) significantly improved the survival rates of *E. coli* cells under either proton or carbon irradiation compared to MG1655^TR^ Red^−^ (*p* < 0.05 for proton irradiation, *p* < 0.01 for carbon irradiation), as shown in Figure 3B. Mutation rates with the expression of the recombinase were significantly decreased under both proton and carbon irradiation (in all cases, *p* < 0.01) (Figure 3C). We further collected Amp^R^ clones to analyze the activation pattern of the *amp* gene, and the results showed that for the MG1655^TR^ Red^−^ strains, 72 Amp^R^ clones were analyzed in the CT group, and 94.4% of the Amp^R^ clones were classified as type I, 4.2% as type II, and 1.4% as type III. In the proton irradiation group, 48.3% (29/60) were type I, 50% (30/60) were type II, and 1.7% (1/60) were type III. In the carbon irradiation group, the proportion of type I was 28.1% (16/57), 64.9% (37/57) were type II, and the rest (7%) were type III. However, for the MG1655^TR^ Red^+^ strains, the CT group exhibited a slight increase in type II (5.8%, 7/120) and a decrease in type I (93.3%, 112/120), yet no significant difference was observed compared to CT group of MG1655^TR^ Red^−^ strains. Particle irradiation, either by 100 Gy proton or carbon radiation, could notably increase the proportion of type II, from 50% to 78.1% in the proton irradiation group and from 64.9% to 78.9% in the carbon irradiation group (*p* < 0.01, in both cases). Thus, the participation of recombinase could significantly stimulate the enhancement of RMD frequency under particle irradiation. However, no significant difference was observed between proton and carbon radiation when analyzing the frequency of RMD rearrangement, thereby suggesting the existence of a threshold for RMD rearrangement. Nonetheless, carbon irradiation significantly promoted the occurrence of mutation type III (Figure 3D). These results suggest a stimulative effect of exogenous recombination, which is initiated by particle irradiation, on RMD rearrangement. 

### 3.4. Role of the RecA-Dependent Pathway on Particle Irradiation-Induced RMD Events

In *E. coli*, RMD occurs via both *RecA*-dependent and *RecA*-independent mechanisms. In the *RecA*-dependent pathway, DSB repair is initiated by the *RecBCD* enzyme; *RecBCD* resects DSB ends, degrading them into ssDNA, and loads the recombinase *RecA* to promote strand exchange [22]. After confirming that exogenous λ-Red recombinase was involved in RMD, we further determined the involvement of endogenous recombinase *RecA* in the induction of RMD following particle irradiation. Here, Δ*recA* and Δ*recB* mutants were adopted by knocking out the *RecA* or *RecB* genes. The results showed that Δ*recA* and Δ*recB* promoted a significant decrease in the survival rate of *E. coli* after proton or carbon irradiation, whereby only 7% and 6% of the cells could survive under 100 Gy proton irradiation for Δ*recA* and Δ*recB,* respectively, with survival rates of 5% and 4% for cells undergoing carbon irradiation (in all cases, *p* < 0.01, compared to the wild type), as shown in Figure 4A. Additionally, analysis of the mutation rates showed that mutation rates in Δ*recA* and Δ*recB* mutants were significantly reduced by one order of magnitude compared to that in wild-type (Figure 4B). Furthermore, the mutation rate of Δ*recA* mutants did not increase following particle irradiation; on the contrary, proton irradiation led to a reduction in mutation rate (*p* < 0.01, compared to CT). For the Δ*recB* mutant, the background mutation rate is as low as in Δ*recA*; however, the difference is that particle irradiation resulted in a significant LET-dependent increase compared to the control (*p* < 0.05 for proton irradiation, *p* < 0.01 for carbon irradiation) (Figure 4B). By combing the extremely low survival rate, it is possible that an overkill effect might play an important role in the low mutation rates observed in the Δ*recA* and Δ*recB* mutants. Next, we analyzed 60 Amp^R^ clones which had been generated by spontaneous mutations in Δ*recA* and found that 96.6% of Amp^R^ clones in the CT group were classified as type I, 1.7% as type II, and 1.7% as type III, which did not significantly differ from the wild-type CT group. For Δ*recA,* a total of 25 Amp^R^ clones generated by 100 Gy proton irradiation and 25 Amp^R^ clones generated by 100 Gy carbon irradiation showed a slight enhancement in type II compared to the CT group—from 1.7% to 4% for proton irradiation and also to 4% for carbon irradiation. However, this degree of enhancement was much lower than that in the wild-type group (from 3% to 51.9% for proton and to 66% for carbon irradiation in the wild-type group) and exhibited significant differences (in both cases, *p* < 0.01, compared to its corresponding wild-type group) (Figure 4C). These results suggested that enhancements in RMD rearrangement induced by particle irradiation are *RecA*-dependent. In addition, particle irradiation could cause an enhancement in RMD frequency in Δ*recB* either for proton irradiation or carbon irradiation (in both cases, *p* < 0.01, compared to the corresponding CT group)*,* yet the degree of enhancement was much lower than that in the wild-type group (in both cases, *p* < 0.01, compared to its corresponding wild-type group), which further supported the role of *RecA* in enhancing RMD rearrangement. In addition, under the circumstance of *RecA* and *RecB* mutation, the frequency of type III induced by particle radiation significantly increased, which suggested that the absence of a recombination mechanism might lead to the activation of other repair pathways such as alternative end joining, which is more prone to structural mutations such as larger fragment deletion and translocation.

## 4. Discussion

Particle irradiation can cause multiple types of DNA damage, among which DSB is the most serious form. However, cells have also evolved a variety of repair pathways to deal with DSBs, among which non-homologous end joining repair (NHEJ) and homologous recombination (HR) repair are the most important DSB repair pathways [23]. NHEJ, which contains two subpathways (canonical NHEJ (c-NHEJ) and alternative end joining (a-EJ)), religates broken ends without requiring a template [24]. Homologous recombination (HR), includes gene conversion, break-induced replication, and single-strand annealing, and uses a homologous donor for repair, which can potentially lead to unexpected rearrangements, such as RMD [25]. RMD is mainly caused by DSBs; site-specific enzyme digestion methods (such as *IsceI* endonuclease) or the CRISPR/Cas system are commonly used to create DSBs [14,16]. The advantage of the aforementioned methods is that they can generate single or multiple DSBs at any specific site; in contrast, ionizing radiation (IR) inflicts a wide spectrum of DNA lesions, including DSBs, that are randomly dispersed throughout the genome and pose greater challenges to DNA damage repair [26,27,28,29]. The most well-accepted characteristic quality parameter of IR is linear energy transfer (LET), with values varying from ~1 keV/µm (low-LET: X, γ-rays) to higher (~10 keV/μm for protons or 10–100 keV/µm for carbon ions) and very high values (>100 keV/µm for alpha particles or heavy-charged particles [30]). The signature of high-LET radiation comprises closely spaced DNA lesions that form a complex cluster of DNA damage, including strand breaks, oxidatively generated base damage, and abasic sites within one or two DNA helical turns [31,32]. However, repairing this complex DNA damage has always been challenging and poorly understood. Therefore, this study focused on RMD rearrangement caused by complex DNA damage following particle irradiation with the aim of providing some clues for elucidating the repair process and possible outcomes of complex DNA damage. First, a reporter strain was used in which *amp* gene activation regulated by *lacI* repeat sequences. Activation of the *amp* gene required either *lacO* inactivation or the simultaneous inactivation of the two *lacI* genes. According to the integrity of the two *lacI* repeats, simultaneous inactivation of two *lacI* repeats was classified into three types. Here, the focus was on type II, in which RMD rearrangement occurred between the two *lacI* genes. Our results demonstrated that particle irradiation could significantly activate the *amp* gene, which was mainly due to an increase in type II, i.e., the enhancement of RMD. 

In this study, the frequency of spontaneous RMD rates was at a relatively low level, approximately 10^−7^ (Figure 1C,D), which was not consistent with previous works, which demonstrated a background RMD rate of 10^−5^–10^−4^ [6]. The special requirement of *amp* activation in the present experimental system might provide an explanation for this difference; the necessity of *amp* activation was not only due to the occurrence of RMD, but was also accompanied by the loss-of-function of residual *lacI* or *lacO,* which indicated that more than one DNA damage event would be generated at the *lacI* site and nearby if the *amp* gene was activated. Therefore, unlike other experimental systems, only the generation of RMD initiated by single DSB could be detected. Generally, IR induces its major lethal or mutagenic effects not solely by the presence of simple DNA damage, like single-strand breaks (SSBs), base lesions, and even a DSB, but primarily by the induction of clustered spatiotemporal damage to DNA, especially following high-LET radiation [30]. This difference in RMD induction was considered to be the main advantage of our experimental system, which was suitable for better understanding RMD induced by complex DNA damage generated by high-LET particle irradiation. Meanwhile, the enhanced frequency of RMD rearrangement exhibited an LET-dependent manner (Figure 1D); the proportion and complexity of clustered DNA lesions increase with the increase in LET [32]. Considering the difference in LET between proton and carbon radiation, high-LET carbon irradiation is more complex in inducing complex DNA damage forms which lead to a significant increase in type II and type III Amp^R^ clones. From the mutation spectrum of type I and type II, especially the symmetrical distribution of point mutations that occur on the *lacI* gene, it can be inferred that some non-DSB damage accumulated in or around the repeat sequence, leaving mutations in the repeat sequence during the repair process. Then, in the later stage of using HR to repair DSB, these mutations would be left in the *lacI* gene after RMD or copied to another repeat sequence through gene conversion [33]. Although most mutations in type II and type I were base pair substitutions (Table 1, Table 2 and Table 3), which seemingly contradicts the traditional concept that high-LET radiation can lead to complex mutation forms, such as larger fragment insertions, deletions or translocations, this is mainly due to our classification of Amp^R^ activation types; complex translocation or structural variation were more likely to be generated in type III.

The rate of DNA recombination rearrangement between repeats is linearly dependent on substrate length [34]. Long DNA repeats, of at least 100 nucletotides, can lead to intrachromosomal rearrangements by acting as substrates for the bacteria recombination machinery [35]. In this study, the length of *lacI* was 1212 bp, which facilitated intermolecular recombination. However, the existence of the exotic plasmid-based homologous *lacI* sequence with different lengths and compositions exhibited various impacts on the frequency of RMD rearrangement between the two intramolecular *lacI* repeats. In terms of the aspect of the Amp^R^ activation rate, a sequence that is homologous upstream of *lacI* has a stronger ability to suppress Amp^R^ activation, whereas an homologous sequence downstream of *lacI* has a weaker ability to promote Amp^R^ activation (ph1sgRNA versus ph2sgRNA versus ph3sgRNA); the longer the length of the homologous sequence, the stronger the ability to suppress Amp^R^ activation (ph4sgRNA versus ph1sgRNA) (Figure 2C). Together, the analysis of the proportion of mutation types revealed that the enhancement of RMD was suppressed, especially for ph4sgRNA, which has a fragment homologous with the complete *lacI* sequence, indicating that the exogenous homologous fragment competitively provided more complete templates, which increased the opportunity to repair DNA damage in the genome using homologous fragments from the plasmids and reduced the probability of RMD occurring between repetitive fragments in the genome.

RMD involves homologous recombination (HR) between two repeat elements. The core step in HR is homology sampling, which is mediated by a conserved RecA-ssDNA nucleoprotein filament that registers an intact identical dsDNA molecule [1]. Recombineering is a simple and efficient way to engineer DNA molecules in vivo by utilizing homologous recombination mechanisms. In *Escherichia coli*, the bacteriophage λ-Red homologous recombination system has been extensively used for genetic modification [36,37]. In this study, the plasmid-based λ-Red recombineering system was adopted to create the reporter strains (MG1655^TR^) and also as an exogenous recombinase (MG1655^TR^ Red^+^), in which the λ-Red recombinase gene was cloned into temperature-sensitive plasmids. Exogenous expression of λ-Red recombinase could improve the usage of homologous recombination repair, thereby leading to an increased survival rate, decreased mutation rate, and enhanced RMD rearrangement (Figure 3B). In contrast, by knocking out the *RecA* gene, the cell survival rate decreased significantly (Figure 4B) and the frequency of RMD was not enhanced under particle irradiation (Figure 4C) due to pronounced recombination and DNA repair deficiency, which strongly supports the conclusion that a *RecA*-dependent HR mechanism is involved in complex DSB repair. While knocking out the *RecB* gene could not suppress the enhancement of RMD induced by particle irradiation, the main reason might be that in *E. coli* cells deficient for RecBCD function (i.e., in *RecB* or/and *recC* null mutants), the RecF pathway may also take over recombination processes that involve dsDNA ends [38,39].

## 5. Conclusions

Overall, using a reporter line of *lacI*-repeat-controlled *amp* activation and taking advantage of complex DSB generation by particle irradiation, this study clearly demonstrated the influence of DSBs complexity on occurrence of RMD rearrangement and the regulation of intermolecular homology and exogenous recombinase. Meanwhile, the *RecA*-dependent pathway was suggested to be involved in the enhancement of RMD under particle irradiation. These results could broaden our understanding of RMD generation and help assess the repair process of complex DSBs located around repeat sequences.

## Figures and Tables

**Figure 1 biology-12-01406-f001:**
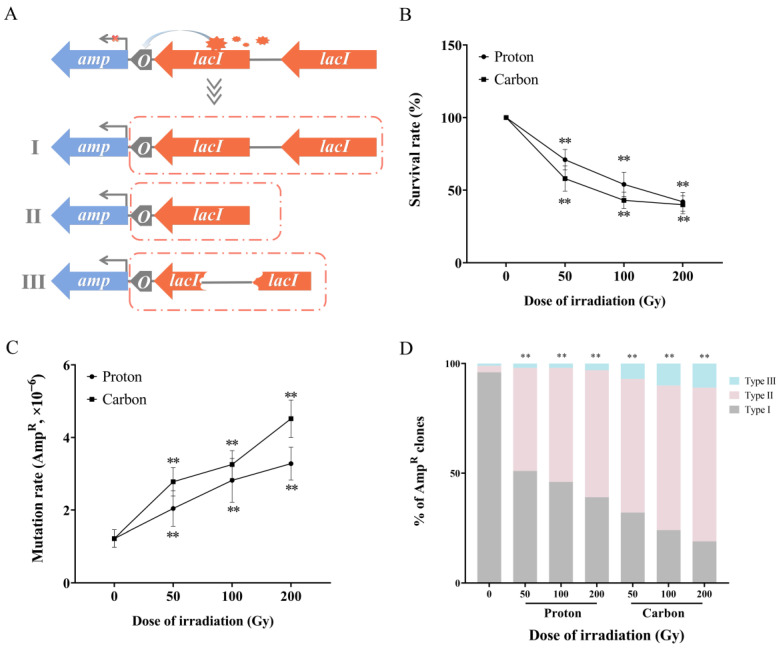
Particle irradiation enhanced the proportion of RMD rearrangement in Amp^R^ activation clones in *Escherichia coli*. (**A**) Schematic diagram of the reporter system, in which the *amp* gene is activated through three types of patterns based on the integrity of two *lacI* repeats: type I is that both *lacI* genes maintain structural integrity together with synchronous point mutations on the two *lacI* or/and on *lacO*; type II is that only one *lacI* remained due to RMD rearrangement together with point mutation on *lacI* or/and on *lacO*; type III is large fragment deletion or complex structural variation occurring on *lacI*. Dashed box represents the loss of function of *lacI* or/and *lacO*. (**B**) Survival rates of *E. coli* cells under particle irradiation (*n* = 5; **, *p* < 0.01). (**C**) Mutation rates of Amp resistance (Amp^R^) induced by the indicated irradiation types (*n* = 5; **, *p* < 0.01). (**D**) Proportional distribution of the three types of Amp activation patterns in Amp^R^ clones under particle irradiation, (**, *p* < 0.01, doses versus 0 Gy), 0 Gy (*n* = 100), 50 Gy proton (*n* = 118), 100 Gy proton (*n* = 152), 200 Gy proton (*n* = 78), 50 Gy carbon (*n* = 188), 100 Gy carbon (*n* = 209), 200 Gy carbon (*n* = 158).

**Figure 2 biology-12-01406-f002:**
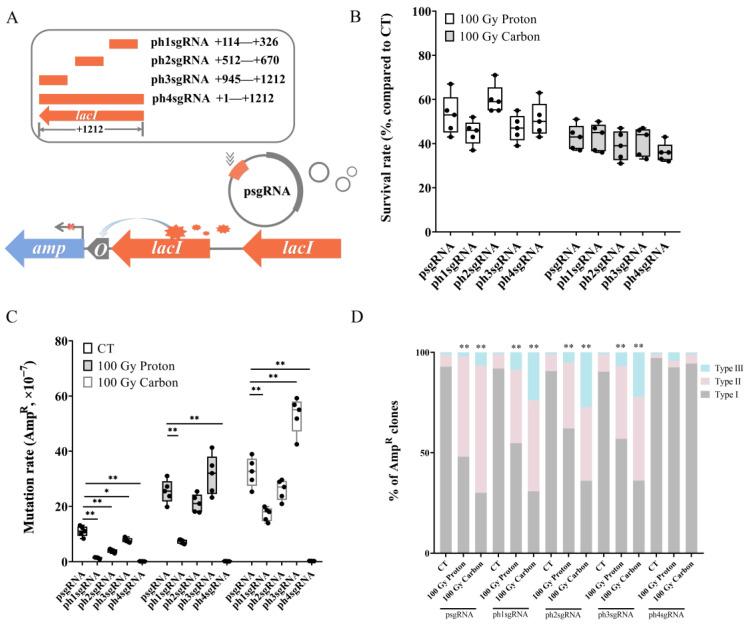
Suppressive effect of intermolecular homology on RMD frequency. (**A**) Schematic diagram of the MG1655^TR^ transfected with plasmids containing various length and constitution of sequences homologous with *lacI*. (**B**) Survival rates of *E. coli* cells under particle irradiation (*n* = 5); (**C**) Mutation rates of Amp resistance (Amp^R^) induced by the indicated irradiation types (*n* = 5; *, *p* < 0.05, **, *p* < 0.01); (**D**) Proportional distribution of the three types of Amp activation patterns in Amp^R^ clones in cells containing various plasmids under particle irradiation, (**, *p* < 0.01, particle irradiation versus its relative CT), CT (*n* = 84), 100 Gy proton (*n* = 50), 100 Gy carbon (*n* = 60) for psgRNA; CT (*n* = 114), 100 Gy proton (*n* = 116), 100 Gy carbon (*n* = 144) for ph1sgRNA; CT (*n* = 96), 100 Gy proton (*n* = 116), 100 Gy carbon (*n* = 139) for ph2sgRNA; CT (*n* = 111), 100 Gy proton (*n* = 126), 100 Gy carbon (*n* = 130) for ph3sgRNA; CT (*n* = 106), 100 Gy proton (*n* = 120), 100 Gy carbon (*n* = 106) for ph4sgRNA. Black solid cycle in each column represents five independent data.

**Figure 3 biology-12-01406-f003:**
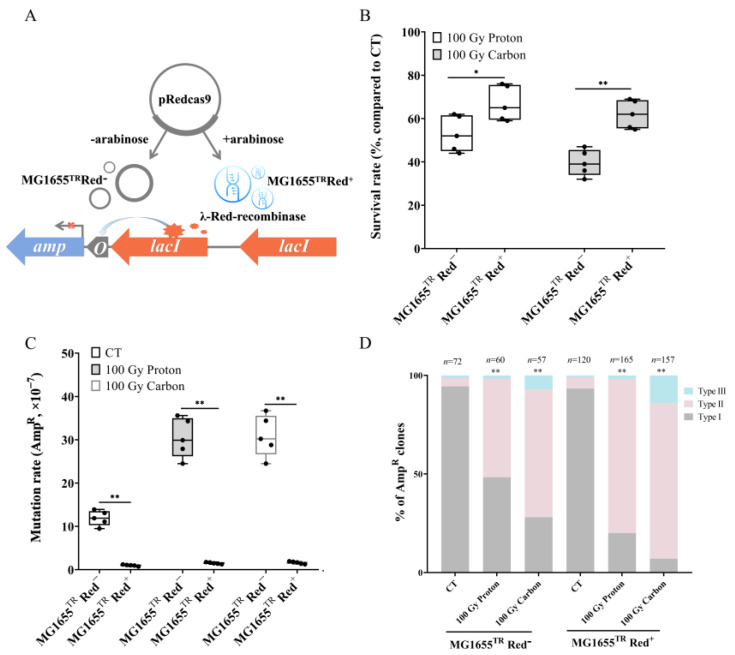
Stimulative effect of exogenous recombinase on RMD induced by particle irradiation. (**A**) Schematic diagram of MG1655^TR^ transfected with pRedCas9 plasmid. (**B**) Survival rates in MG1655^TR^ Red^−^ or MG1655^TR^ Red^+^ strains under particle irradiation (*n* = 5; *, *p* < 0.05; **, *p* < 0.01). (**C**) Mutation rates of Amp resistance (Amp^R^) in MG1655^TR^ Red^−^ or MG1655^TR^ Red^+^ strains induced by the indicated irradiation types (*n* = 5; **, *p* < 0.01). (**D**) Proportional distribution of the three types of Amp activation patterns in Amp^R^ clones collected from MG1655^TR^ Red^−^ and MG1655^TR^ Red^+^ under particle irradiation (**, *p* < 0.01, particle irradiation versus its relative CT); CT (*n* = 72), 100 Gy proton (*n* = 60), 100 Gy carbon (*n* = 57) for MG1655^TR^ Red^−^; CT (*n* = 120), 100 Gy proton (*n* = 165), 100 Gy carbon (*n* = 157) for MG1655^TR^ Red^+^. Black solid cycle in each column represents five independent data.

**Figure 4 biology-12-01406-f004:**
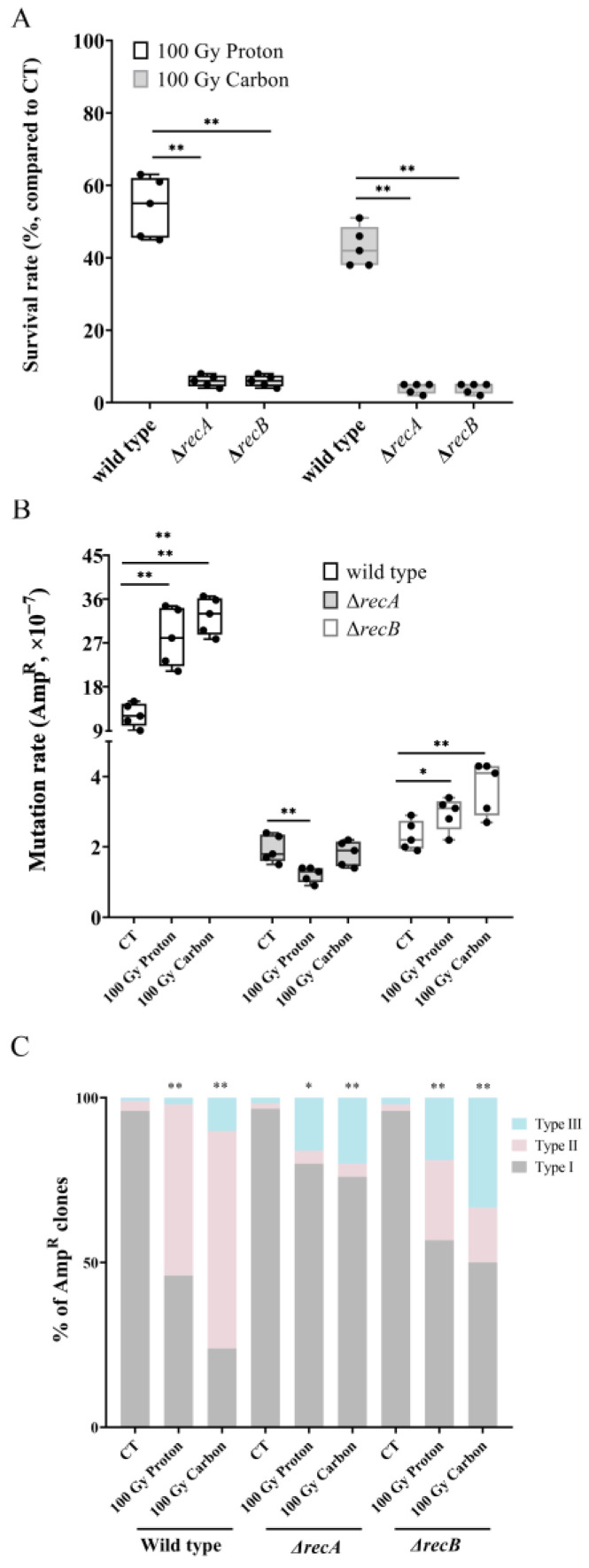
Role of *RecA*-dependent pathway on particle irradiation-induced RMD event. (**A**) Survival rates of Δ*recA* and Δ*recB* mutants under particle irradiation (*n* = 5; **, *p* < 0.01). (**B**) Mutation rates of Amp resistance (Amp^R^) in Δ*recA* and Δ*recB* mutants induced by particle irradiation (*n* = 5; *, *p* < 0.05; **, *p* < 0.01). (**C**) Proportional distribution of the three types of Amp activation patterns in Amp^R^ clones collected from wild-type, Δ*recA,* and Δ*recB* mutants under particle irradiation (**, *p* < 0.01, particle irradiation versus its relative CT); CT (*n* = 100), 100 Gy proton (*n* = 152), 100 Gy carbon (*n* = 209) for wild type; CT (*n* = 25), 100 Gy proton (*n* = 26), 100 Gy carbon (*n* = 50) for Δ*recA*; CT (*n* = 50), 100 Gy proton (*n* = 37), 100 Gy carbon (*n* = 18) for Δ*recB*. Black solid cycle in each column represents five independent data.

**Table 1 biology-12-01406-t001:** Type II mutation spectrum of *lacO* and *lacI* genes in Amp resistance (Amp^R^) clones collected from 0 Gy group. “→” represents substitution; “+” represents insertion.

	*lacO*		*lacI*	
No.	Site	Mutation	Site	Mutation
1	+4	C→T	-	-
2	+4	C→T	-	-
3	-	-	+568	T+

**Table 2 biology-12-01406-t002:** Type II mutation spectrum of *lacO* and *lacI* genes in Amp resistance (Amp^R^) clones collected from 100 Gy proton irradiation group. “→” represents substitution; “+” represents insertion.

	*lacO*		*lacI*			*lacO*		*lacI*	
No.	Site	Mutation	Site	Mutation	No.	Site	Mutation	Site	Mutation
1	+4	C→T	-	-	30	+4	C→T	-	-
2	+4	C→T	-	-	31	+4	C→T	-	-
3	+4	C→T	-	-	32	+4	C→G	-	-
4	+4	C→T	-	-	33	+4	C→G	-	-
5	+4	C→T	-	-	34	+4	C→T	-	-
6	+4	C→T	-	-	35	+4	C→G	-	-
7	+4	C→T	-	-	36	+4	C→T	-	-
8	+4	C→T	-	-	37	+4	C→G	-	-
9	+4	C→T	-	-	38	+4	C→G	-	-
10	+4	C→T	-	-	39	+6	C→T	-	-
11	+4	C→T	-	-	40	+8	C→A	-	-
12	+4	C→T	-	-	41	+9	C→T	-	-
13	+4	C→T	-	-	42	+9	C→T	-	-
14	+4	C→T	-	-	43	+9	C→T	-	-
15	+4	C→T	-	-	44	+9	G→A	-	-
16	+4	C→T	-	-	45	+9	G→A	-	-
17	+4	C→T	-	-	46	-	-	+571	G→A
18	+4	C→T	-	-	47	-	-	+580	C+
19	+4	C→T	-	-	48	-	-	+592	G+
20	+4	C→T	-	-	49	-	-	+593	A+
21	+4	C→T	-	-	50	-	-	+595	G→T
22	+4	C→T	-	-	51	-	-	+595	C+
23	+4	C→T	-	-	52	-	-	+599	C+
24	+4	C→T	-	-	53	-	-	+607	T→G
25	+4	C→T	-	-	54	-	-	+628	T+
26	+4	C→T	-	-	55	-	-	+635	G+
27	+4	C→T	-	-	56	-	-	+672	G+
28	+4	C→T	-	-	57	-	-	+755	G→A
29	+4	C→T	-	-	58	-	-	+836	G→T

**Table 3 biology-12-01406-t003:** Type II mutation spectrum of *lacO* and *lacI* genes in Amp resistance (Amp^R^) clones collected from 100 Gy carbon irradiation group. “→” represents substitution; “+” represents insertion; “−” represents deletion.

	*lacO*		*lacI*			*lacO*		*lacI*	
No.	Site	Mutation	Site	Mutation	No.	Site	Mutation	Site	Mutation
1	+4	C→T	-	-	54	-	-	+572	G+
2	+4	C→T	-	-	55	-	-	+577	A+
3	+4	C→T	-	-	56	-	-	+582	G→T
4	+4	C→T	-	-	57	-	-	+599	G→A
5	+4	C→T	-	-	58	-	-	+603	G+
6	+4	C→T	-	-	59	-	-	+603	GCCA−
7	+4	C→T	-	-	60	-	-	+604	GCCA+
8	+4	C→T	-	-	61	-	-	+604	GCCA+
9	+4	C→T	-	-	62	-	-	+604	GCCA−
10	+4	C→T	-	-	63	-	-	+604	GCCA+
11	+4	C→T	-	-	64	-	-	+643	C+
12	+4	C→T	-	-	65	-	-	+653	G→C
13	+4	C→G	-	-	66	-	-	+653	G→C
14	+4	C→T	-	-	67	-	-	+659	C→T
15	+4	C→T	-	-	68	-	-	+663	C→G
16	+4	C→T	-	-	69	-	-	+667	7 bp+
17	+4	C→T	-	-	70	-	-	+671	G→A
18	+4	C→T	-	-	71	-	-	+674	C+
19	+4	C→T	-	-	72	-	-	+696	G→C
20	+4	C→T	-	-	73	-	-	+696	G→C
21	+4	C→T	-	-	74	-	-	+710	G−
22	+4	C→T	-	-	75	-	-	+710	G→A
23	+4	C→T	-	-	76	-	-	+711	C→A
24	+4	C→T	-	-	77	-	-	+718	G→C
25	+4	C→T	-	-	78	-	-	+720	A+
26	+4	C→T	-	-	79	-	-	+726	G→A
27	+4	C→T	-	-	80	-	-	+735	G+
28	+4	C→T	-	-	81	-	-	+735	T→C
29	+4	C→T	-	-	82	-	-	+735	G+
30	+4	C→T	-	-	83	-	-	+735	G+
31	+4	C→T	-	-	84	-	-	+750	G→C
32	+4	C→T	-	-	85	-	-	+758	201 bp−
33	+4	C→T	-	-	86	-	-	+771	G→A
34	+4	C→T	-	-	87	-	-	+783	G→A
35	+4	C→T	-	-	88	-	-	+802	11 bp−
36	+4	C→G	-	-	89	-	-	+803	G→A
37	+4	C→T	-	-	90	-	-	+821	GTCGT−
38	+4	C→T	-	-	91	-	-	+848	G→A
39	+4	C→T	-	-	92	-	-	+849	G→A
40	+4	C→T	-	-	93	-	-	+879	G→A
41	+4	C→T	-	-	94	-	-	+885	G→C
42	+4	C→T	-	-	95	-	-	+899	G→T
43	+5	A→G	-	-	96	-	-	+905	G→C
44	+5	A→G	-	-	97	-	-	+922	G→A
45	+9	G→A	-	-	98	-	-	+922	G→T
46	+9	G→T	-	-	99	-	-	+941	49 bp−
47	+9	G→T	-	-	100	-	-	+956	C→A
48	+9	G→T	-	-	101	-	-	+957	8 bp+
49	+9	G→T	-	-	102	-	-	+962	G→A
50	+9	C→T	-	-	103	-	-	+966	G→A
51	+10	C→A	-	-	104	-	-	+970	A→G
52	+10	C→T	-	-	105	-	-	+1001	CG→AT
53	-	-	+572	G+	106	-	-	+1018	G+

## Data Availability

All data associated with this study are available within the manuscript and Appendix A.

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
