# Peer review of "Enhancement of Repeat-Mediated Deletion Rearrangement Induced by Particle Irradiation in a RecA-Dependent Manner in Escherichia coli"

_biology, 2023, doi:10.3390/biology12111406_

Round 1

Reviewer 1 Report

Comments and Suggestions for Authors

The study by Hou et al. explores the repeat-mediated deletion (RMD) process during the generation of complex double-strand breaks (DSBs) through particle irradiation. They used the lacI-repeats-controlled amp activation reporter line to investigate the regulation of intermolecular homology and exogenous recombinase. The research suggests that the RecA-dependent pathway plays a significant role in enhancing RMD during particle irradiation. The data and results presented in the study are well-supported. 

However, including an additional experiment to check the effect of recombinase and RecA pathway on ph4sgRNA with complete lacI homologous sequence to suppress RMD rearrangements induced by particle irradiation would further strengthen the manuscript.

Author Response

Thank you for your positive comments and valuable suggestions. In the result 2 part, we artificially transfected four plasmids containing different length of homology into the cells in which RecA is intact, we think that RecA-dependent homologous repair mediated the recombination between intra- and inter- homologous sequence, but in order to further strength this conclusion, we need to transfected the ph4sgRNA into RecA mutant cells, and we are happy to perform this experiment, but due to the limitation of radiation, in this work, proton and carbon ions are generated by accelerators which are only available at a certain time of year, we also need to apply for the radiation time and submit related experiment design a year in advance, and the following radiation experiment would be arranged in the next year, so we cannot complete this experiment in a short period, we would like to exhibit related result in a corrected version if this work would be accepted to publish.

Reviewer 2 Report

Comments and Suggestions for Authors

First of all I would like to congratulate the authors for their hard work. I have few minor comments that authors can take care of it which are as follows:

1) Mention the annealing temperature for PCR reaction the supplementary data inform of each pair of primer used

2) Mention the concentration of DNA use for PCR reaction. 

Author Response

Thank you for your positive comments, and we have added related information in the supplementary table 2 and the revised manuscript.

Reviewer 3 Report

Comments and Suggestions for Authors

This study deals with repair of irradiation-induced lesions in E. coli, with an emphasis on repeat-mediated deletions (RMD) and point mutations resulting from it. The authors show that RMD occurrence is increased upon irradiation, which depends on RecA protein, as well as on Red recombinase, but is suppressed by intermolecular homologous sequence.

In my opinion, the authors miss one important aspect: namely, while the RecA is indeed required for DSB repair that could lead to RMD, it is also required for induction of SOS response to irradiation. So, RecA  mutant is also SOS def, which has pleiotropic effects on E. coli physiology, including mutagenesis, DNA repair,  cell division etc. I suggest authors differentiate the role of SOS induction and DSB repair (some of that is already apparent in different effects of recB and recA mutations on RMD profile, as both mutations abolish DSB repair while only the latter inhibits SOS regulon) by for instance using lexA3 (SOS-) mutant containing recAo281 operator mutation that enables SOS-independent constitutive RecA overexpression (eg. G3, 7, 2017, doi.org/10.1534/g3.117.043521). Also, the authors could check the origin of the observed mutagenesis (ie its dependence on PolV - UmuCD, which is  also part of SOS).

Moreover, it would be interesting to see the effect of Rad recombinase in a recA deficient cells. Also, since the clustered damage arises from particle irradiation (but it is the case with gamma rays too, maybe in more moderate level) the effect of proteins that process those lesions should be characterized (especially considering that RecBCD enzyme binds nearly blunt DNA ends), for instance single-strand dependent exonucleases (eg RecJ, ExoI etc).

For this study, I think the resolution of the SOS role should be done, the other suggestions are just ideas for the future work.

Comments on the Quality of English Language

The English usage is correct

Author Response

 Response: Thank you for your constructive suggestions. Yes, we totally agree with you that RecA is not only involved in the DSB repair, but also in the SOS response, our results also showed that mutant in RecA caused the lower survival rate and RMD occurrence, so we inferred that the enhanced RMD induced by particle irradiation is RecA-dependent, and we think this conclusion might be not affected if the related SOS response was involved. As we also know, radiation could also induce the SOS response, and in this work, we are not pay much attention on this inner complex response involved, we are focusing more on the phenomenon that particle irradiation could enhance the RMD occurrence. Of course, the underlying mechanism is also important, as you mentioned that SOS response might be involved, and we also have great interest to construct such mutants like lexA3 (SOS-) mutant containing recAo281 operator mutation and perform related experiments to elucidate this issue, but due to the limitation of radiation, in this work, proton and carbon ions are generated by accelerators which are only available at a certain time of year, we need to apply for the radiation time a year in advance, at the same time the deadline for submitting the revised manuscript is 18 Nov, so we cannot complete this series of experiments in this short time, but we are happy to preform those experiment in our following work.

The another interest issue is the effect of Red recombinase in a recA deficient cells, theoretically speaking, Red recombinase might compensate the deficient of RecA, for their similar function in DNA resection and recombination, but it needs more experiment to confirm, and we like to perform these experiment together with that mentioned before. We are hopeful to finish the new work and invite you to review it.

Thank you again for your valuable suggestions.

Round 2

Reviewer 3 Report

Comments and Suggestions for Authors

Although I still consider the SOS role in the assayed phenomena an important point, I appreciate the problems and limitations you have in the experimental work. Therefore, I consider the manuscript suitable for publication in Biology in its current form.